# Insights on the Mechanical Properties of SARS-CoV-2 Particles and the Effects of the Photosensitizer Hypericin

**DOI:** 10.3390/ijms25168724

**Published:** 2024-08-10

**Authors:** Matteo Mariangeli, Ana Moreno, Pietro Delcanale, Stefania Abbruzzetti, Alberto Diaspro, Cristiano Viappiani, Paolo Bianchini

**Affiliations:** 1Dipartimento di Scienze Matematiche, Fisiche e Informatiche, Università di Parma, 43124 Parma, Italy; matteo.mariangeli@iit.it (M.M.); pietro.delcanale@unipr.it (P.D.); stefania.abbruzzetti@unipr.it (S.A.); 2Nanoscopy and NIC@IIT, Center for Human Technology, Istituto Italiano di Tecnologia, 16152 Genova, Italy; alberto.diaspro@iit.it; 3Istituto Zooprofilattico Sperimentale della Lombardia e dell’Emilia-Romagna, 25124 Brescia, Italy; anamaria.morenomartin@izsler.it; 4DIFILAB, Dipartimento di Fisica, Università di Genova, 16146 Genova, Italy

**Keywords:** SARS-CoV-2, hypericin, photodynamic therapy (PDT), enveloped viruses, Atomic Force Microscopy (AFM), nanomechanical properties, AFM-fluorescence correlative microscopy

## Abstract

SARS-CoV-2 is a highly pathogenic virus responsible for the COVID-19 disease. It belongs to the Coronaviridae family, characterized by a phospholipid envelope, which is crucial for viral entry and replication in host cells. Hypericin, a lipophilic, naturally occurring photosensitizer, was reported to effectively inactivate enveloped viruses, including SARS-CoV-2, upon light irradiation. In addition to its photodynamic activity, Hyp was found to exert an antiviral action also in the dark. This study explores the mechanical properties of heat-inactivated SARS-CoV-2 viral particles using Atomic Force Microscopy (AFM). Results reveal a flexible structure under external stress, potentially contributing to the virus pathogenicity. Although the fixation protocol causes damage to some particles, correlation with fluorescence demonstrates colocalization of partially degraded virions with their genome. The impact of hypericin on the mechanical properties of the virus was assessed and found particularly relevant in dark conditions. These preliminary results suggest that hypericin can affect the mechanical properties of the viral envelope, an effect that warrants further investigation in the context of antiviral therapies.

## 1. Introduction

Coronaviruses are a diverse family of viruses capable of infecting various animals, including humans. Characterized by their specific nucleic acid composition, an external lipid envelope, and unique morphology, they are single-stranded positive-sense RNA viruses [1,2] with sizes ranging from 60 nm to 140 nm. The spike proteins on the virus membrane give them a distinct “crown-like” appearance under an electron microscope [3], hence the origin of their name. The viral envelope plays a pivotal role in the infection process of these pathogens, facilitating the initial attachment to the host cell and mediating the fusion process, thereby enabling the viral genome to penetrate the host cell [4,5]. Research in this domain is constantly advancing, offering a new understanding of the mechanisms behind viral cell infection and identifying possible targets for the development of antiviral treatments. In particular, compounds that are able to affect the mechanical properties of the viral envelope are of special interest, since they introduce a new concept in antiviral research, opening perspectives for developing broad-spectrum antivirals [6].

Hypericin (Hyp), a natural pigment derived from the common St. John’s Wort plant is a promising candidate for antiviral therapy, being effective in inactivating enveloped viruses [7,8,9]. Its lipophilic nature allows it to bind with high-affinity phospholipidic membranes [10,11,12] and protein hydrophobic pockets [13]. As a photosensitizer [14], Hyp triggers the production of reactive oxygen species (ROS), like singlet oxygen (^1^O_2_). ROS are harmful to biological systems and can be exploited to kill cancer cells or pathogens [15,16,17]. Hence, the primary antiviral effect of Hyp upon light irradiation is due to its photodynamic activity. Recent studies have shown that Hyp retains antiviral properties even in the absence of light, suggesting a multimodal activity [6]. The binding of Hyp to the viral membrane appears to be a fundamental event to elicit its antiviral effects, which disappear in viruses devoid of a viral envelope [7,18,19]. In this work, we investigate the effects of Hyp on the mechanical properties of the viral envelope, in the dark and under visible light irradiation.

There is increasing evidence that enveloped viruses possess structural adaptability, relevant from an evolutionary perspective. Virions can be exposed to physical forces such as shear stress in viscous fluids and osmotic or hydrostatic pressure, which can deform or fracture them. The mechanical characteristics of viral particles likely evolved in response to such selection pressure, by aiding viruses to withstand or even leverage these forces [20]. Studies on viruses like Murine Leukemia Virus and HIV-1 reveal that immature virions are much stiffer than their more infective mature counterparts, highlighting the metastable nature of their mechanical properties [21,22,23]. Atomic Force Microscopy (AFM) has been pivotal in examining the morphology and mechanical properties of virus shells in a liquid environment [24]. Recent research considered the connection of viral mechanical properties with the dense genome packing and the viral maturation process, employing techniques like mechanical fatigue and nanoindentation [25,26,27].

Taking these factors into account, we focused our study on the nanomechanical properties and imaging of single SARS-CoV-2 viral particles using AFM to uncover key characteristics.

The chosen AFM modalities for this investigation were Jumping mode [28,29,30] and QI^TM^ (Quantitative Imaging) [31]. In Jumping mode AFM, the tip touches the sample surface for a very short time and then it is retracted. This process is repeated as the cantilever scans across the sample surface pixel by pixel. The cantilever “jumps” on the surface, spending most of its time away from the sample. The force exerted is precisely controlled and minimized, which is crucial for studying soft and easily detachable biological samples. Moreover, in addition to setting the force applied at each point, there is the possibility to define a maximum force threshold. If this threshold is reached (due to debris contacting the tip, surface irregularities, etc.), the tip quickly retracts to avoid any damage. QI^TM^ works in a similar way, even though it is not possible to set the maximum bearable force, like in Jumping mode. In QI^TM^ it is possible to record force–distance (F–D) curves at each pixel of the image, giving information about the local interaction between the tip and the sample. The advantage of both modalities is the minimization of the shear lateral forces induced by the tip, allowing the study of easily detachable samples.

Due to safety requirements, working with active SARS-CoV-2 virus was not feasible as it requires a biosafety level 3 (BSL3) conditions. Consequently, we opted for a heat-inactivated virus because chemical inactivation introduces crosslinks that irreversibly alter the virus’s mechanical structure. In addition, heat inactivation can partially degrade the viral RNA and membrane proteins, or affect the virus’s structural integrity, including the envelope, as indicated in previous research [32]. Nevertheless, existing research on heat-inactivated viruses [33] suggests that it is possible to maintain intact viral particles, provided that the sample is properly prepared.

## 2. Results and Discussion

### 2.1. AFM Imaging

The initial experiments focused on assessing the basic morphological features of the viruses, e.g., their shape and height. Typically, we expect to observe round objects, decorated with viral proteins protruding from the surface, consistent with the structure of SARS-CoV-2 [33]. Figure 1 displays a typical Atomic Force Microscopy (AFM) image of an intact viral particle, where remnants of spike proteins can be seen on the round surface (arrows in Figure 1). At a distance from the particle, other irregular formations are visible, that may be identified with debris from damaged particles, such as proteins, RNA, or fragments of the viral envelope, likely a result of the inactivation process. It is improbable that these irregular objects are due to the poly-L-lysine substrate, as it has been verified to be quite flat and smooth, with minimal impurities, typically less than 5 nm in height.

Therefore, while electron microscopy allows imaging of the structural features of the spike glycoproteins on the virus’s surface and determining their height relative to the envelope, AFM imaging of these proteins is challenging. This limitation arises from the imaging technique used, where a scanning tip sweeps over the surface and likely displaces the proteins instead of accurately recording their heights. However, as noted, the general aspects of structures can be observed [34]. Additionally, it is crucial to note that the observed height of the particles reflects not only their inherent structure but also their interaction with the underlying substrate. This aspect is particularly relevant for enveloped viruses, where the envelope imparts additional flexibility to the structure.

The experiment depicted in Figure 2 was designed to evaluate the resistance of samples to repeated imaging, essentially conducting a fatigue test. Fatigue measurements generally refer to the methods and techniques used to quantify the level of fatigue experienced by a material. In the context of virus shells, mechanical fatigue pertains to their ability to withstand a force before structural collapse [35]. The breaking force can provide valuable insights into the uncoating pathway of viruses. Continuous AFM imaging is used to observe any structural changes in the protein shell under repeated low-force load cycles (usually 50–150 pN per pixel), demonstrating the shell’s stability against multiple deformations at such forces.

The typical result of a fatigue experiment is shown in Figure 2, which effectively summarizes the findings from multiple tests we performed. Specifically, Figure 2a presents the initial image, and Figure 2b shows the final one from the fatigue test. These images were taken at a force of 50 pN, and the experiment lasted about 70 min, during which 24 images were captured. Analysis of the images and the line profile shown in Figure 2c indicates that the virus did not completely break. Instead, it appears to have lost some material, as evidenced by a height reduction of about 10 nm. This outcome is particularly significant, as viruses with protein shells usually disassemble more distinctly in such measurements [36]. Conversely, the presence of a phospholipid envelope may confer a degree of pleomorphism to the virions, enabling them to alter their structure under mechanical stress without fully collapsing.

The above behavior of the SARS-CoV-2 under fatigue tests may indicate structural adaptability and inherent softness of this virus. In principle, a quantitative estimate of the observed mechanical behavior of the virion would be provided by Young’s modulus. However, an accurate determination of Young’s modulus would require knowledge of the sample’s thickness [37]. Given the variability in the heights of the viral particles (60–150 nm), a much larger dataset would be necessary than the available one. Therefore, we did not proceed with the determination of the Young’s moduli and exploited a different approach, based on the measurement of the breakthrough force (vide infra).

### 2.2. Correlative AFM-Fluorescence Microscopy

The possibility of performing confocal measurements in the same experimental system gave us the opportunity to obtain some correlative AFM-fluorescence images [38,39,40,41,42,43] as a control of the overall quality of our viral particles. This instance is very important, considering the large amount of damaged virions that we came across with AFM.

Particularly, we stained the viral particles with an RNA-binding fluorophore, Syto13 to check if the genetic material was colocalizing with the viral particles. The AFM microscope from JPK that was available for correlative imaging did not allow any gentler Quantitative Imaging (QI^TM^), therefore tapping mode in liquid [44] was the choice modality. In the tapping mode, the cantilever oscillates at its resonance frequency. As the tip approaches the sample surface, it lightly “taps” the surface at the bottom of its oscillation cycle. This modality is not ideal for our sample, because the lateral forces are not negligible. Anyway, despite experiencing some dragging in the virions, we were able to find some correlations. Figure 3 is an example of correlation between the fluorescence signal coming from the viral RNA (Figure 3a) and the AFM topography map that we found (Figure 3b). Interestingly, the RNA is localized also on partially damaged virions (Figure 3c), showing that viruses that look damaged but not completely disrupted appear to retain their genetic material.

### 2.3. AFM Nanomechanical Properties

Nanoindentation involved recording force–distance (F–D) curves [45] to uncover the mechanical properties of the virus. A linear response corresponds to the elastic behavior of the shell, while a shift to Hertzian behavior occurs if the shell is filled. Exceeding a critical point causes the particle to break (breakthrough force) [46]. Additionally, nanoindentation can reveal potential self-recovery abilities of viral envelopes or protein cages [47].

The breakthrough force reflects the mechanical properties of the viral envelope and is linked to the lateral interactions of phospholipid molecules [48,49], hence it provides a direct measure of the mechanical stability of this viral compartment. Measurements were conducted under three different conditions: without Hyp, with Hyp in the dark, and with Hyp under light irradiation, to observe changes due to lipid assembly modifications or photo-oxidation of the viral envelope.

A typical measurement of SARS-CoV-2 started with imaging of a few particles, to check the presence of possible intact ones (Figure 4a). The chosen virions were there centered in the field of view (e.g., the one indicated by the dashed square), and the F–D curves were acquired, by indenting in correspondence to the midpoint of the particle. The chosen single particles were further subject to breakthrough analysis by collecting a force–distance curve (Figure 4b). The plot shows clear penetration of the external viral envelope at a force of ca. 0.5 nN.

To integrate Hyp effectively into the viral particles, we mixed the photosensitizer and the virus in a vial, then deposited the mixture on the substrate as described in the Materials and Methods section. Although this protocol prevented direct comparisons of the same viral particle before and after Hyp treatment, we should mention that even if Hyp had been applied directly to the sample, it would have been highly unlikely to re-identify and monitor the same virion for further observation.

Admittedly, the measurements presented challenges; clear breakthroughs, that indicate penetration through an intact structure, were not always achieved. Often, the breakthroughs occurred in already damaged virions, which were difficult to distinguish from intact ones in the images, likely due to the inactivation process causing irregularities on the sample. Importantly, the structure of intact particles often broke after several indentation cycles (typically 5–10), consistent with prior findings [33] and highlighting the resilience of the SARS-CoV-2 viral envelope.

We should mention that while the viral envelope is expected to play a major role in adaptability of these viruses, we cannot exclude that the mechanical properties exposed by breakthrough force experiments may reflect other contributions due to, e.g., RNA packing and membrane proteins [25,50,51].

Breakthrough forces determined for SARS-CoV-2 in the absence and in the presence of 1 μM Hyp, either in the dark or under illumination are summarized in Figure 4c and Table 1. When Hyp is added to the viral suspension, a clear two-fold increase in the breakthrough force is observed in the absence of illumination. When particles loaded with Hyp were subject to illumination we found that the breakthrough force was comparable to the one determined for untreated viruses. This finding indicates that, as far as the breakthrough force is concerned, the effects of photooxidation induced by Hyp tend to compensate for the changes in mechanical properties of the envelope brought about by Hyp binding. A molecular explanation for this finding is yet to be understood.

## 3. Materials and Methods

The experiments were conducted using different AFM setups: a Nanotec Cervantes (Nanotec, Madrid, Spain) in AFM Jumping mode modality [29], a JPK NanoWizard III in QI^TM^ modality, and a JPK NanoWizard II in tapping mode(JPK BioAFM, Bruker Nano GmbH, Berlin, Germany). The latter microscope was mounted on a Leica SP5 (Leica Microsystems, Mannheim, Germany) inverted laser scanning microscope. Jumping mode and QI^TM^ are highly advantageous for handling delicate and easily detachable viral particles [52] because they drastically reduce lateral/dragging forces. For all the measurements we opted for the rectangular qp-BioAC cantilevers from NanoAndMore (NanoAndMore gmbh, Wetzlar, Germany), with a nominal spring constant of around 0.1 N/m (0.06–0.18 N/m) and tip radii of less than 10 nm.

### 3.1. Imaging and Fatigue Experiments

As previously mentioned, for the images of the virions and fatigue experiments we used Jumping mode in water and QI^TM^.

#### 3.1.1. Jumping Mode

The sensitivity of the tip was calibrated by performing an F–D curve and fitting the linear part of the curve; such a calibration process resulted in typical sensitivity values of around 6–7 nm/V. Data acquisition and analysis were carried out using the WSxM v4.0 Beta 10.0 free software [53]. Images were captured at a resolution of 128 pixels at scanning rates of 0.7–1.5 Hz, within very small fields of view (200–500 nm) at very low forces (less than 100 pN) to maintain the integrity of the virions. Fatigue measurements were conducted under the same conditions, with forces ranging from 50 to 150 pN.

#### 3.1.2. QI^TM^ Mode

The sensitivity for QI^TM^ measurements was determined in the same way, obtaining the same output. In this case, the spring constant was also experimentally evaluated by performing the contact-free thermal noise calibration method [54]. We found values in the range of 0.1–0.15 N/m, in accordance with the nominal ones.

Data acquisition and analysis were carried out with the JPKSPM software v. 8.0. We typically acquired images at 128 × 128 pixels or 256 × 256 pixels with 0.5–2 μm fields of view. The speed ranged between 3 ms and 6 ms per pixel, with a Z-length of 200 nm.

### 3.2. Nanomechanics

F–D curves at the midpoint of the particle were performed aiming to observe at least one breakthrough event. These curves commenced 40–100 nm away from the particle, with the total z-piezo displacement length varying based on the viral particle’s size, typically falling within the range of 100–160 nm. We captured these curves at resolutions of either 128 or 256 pixels, with speeds ranging between 100–150 nm/s. Following the set of F–D curves, we conducted an image of the viral particle to evaluate its integrity, ensuring no damage or collapse had occurred.

The analysis comprised selecting only the clear breakthrough forces, which accounted for less than 50% of the total acquired curves. Initially, we calibrated the curves in WSxM or JPKSPM to determine the actual tip-sample distance. Subsequently, the breakthrough force values (FB) were plotted in OriginPro (OriginLab corp., Northampton, MA, USA) to analyze their distributions.

For the measurements in light, we employed a source of irradiation focused on the sample. Particularly, we used the ACE 1 Halogen Light Source with EKE Lamp, A20500 (Schott, UK), employing a 475/35 filter. In this scenario, the measured power at the sample was P_490_ = 2.5 mW, and the density was ~1.5 × 10^−2^ W/cm^2^. Measurements upon light exposure started after approximately 15 min of irradiation and were then performed under continuous irradiation. On the contrary, during dark measurements, the lamp, as well as the surrounding lights, were turned off.

### 3.3. Correlative Microscopy

Correlative AFM-fluorescence was performed by means of a JPK NanoWizard II (JPK BioAFM, Bruker Nano GmbH, Berlin, Germany) mounted on a Leica SP5 (Leica Microsystems, Mannheim, Germany) inverted laser scanning microscope. The usual stage of the light microscope was substituted by an AFM-compatible one, also possessing minimal mechanical noise. Everything was placed on an active vibration isolation table.

The modality for AFM imaging was tapping mode in liquid with resonance frequencies around 18–22 kHz and scanning rates of 0.2–0.7 Hz. Imaging at high resolution was very challenging, due to the dragging forces exerted by the tip in this modality, and by the fact that we do not have control over the exerted forces. To improve the stability, we opted for imaging at low resolutions (64 × 64 pixels or 128 × 128 pixels) with less time spent scanning on the virions. All the images were analyzed with the JPKSPM software v. 8.0.

The confocal images were acquired with a supercontinuum laser SuperK extreme (NKT Photonics, Birkerød, Danmark) by selecting λ_exc_ = 488 nm as excitation. A notch filter at 488/10 nm was employed, to eliminate any left shot noise coming from the excitation laser. We used an HCX PL APO CS 100.0 × 1.40 oil objective and detected the fluorescence signal at λ_em_ = 500–570 nm through a Leica HyD detector. The pinhole was set at 0.7 a.u., 1024 × 1024 pixels, scan rate 1000 Hz, 16-line averages, pixel size 30 nm.

Since we were not aware of the correlation ratio, we used 80 nm gold beads as fiducial markers and detected the reflection at λ_exc_ = 633 by means of a PMT detector. This enabled a correct alignment of the AFM and light images. We diluted the gold beads stock 1:100 and deposited a small drop of 50–100 μL over the glass coverslip, already coated with poly-L-lysine; we then let the drop evaporate overnight at 4°. The correlation between the different images requires a fine calibration procedure that is aided by the use of a module on the JPKSPM software v. 8.0, called DirectOverlay. By capturing the images in reflection of the AFM cantilever, the software is able to account for the spherical aberrations introduced by the light microscope; it transforms the confocal image as it was completely planar, in AFM coordinates. Syto13 from ThermoFisher Scientific (Thermo Fisher Scientific Inc., Waltham, Ma, USA) was diluted 1:1000 in PBS buffer at pH = 7.4. We added only a small volume to the glass slide (50–100 μL) once the viral particles were already deposited.

### 3.4. Virus Preparation and Inactivation

The SARS-CoV-2 stock of viral particles was prepared in BSL3 laboratories. The SARS-CoV-2 viral strain (HCoV-19/Italy/310904/46/2020) used in this study was obtained by inoculating infected human biological specimens into Vero E6 cell lines. The SARS-CoV-2 strain was then propagated in the same cell line and incubated at 37 °C with 5% CO2. The viral titer (TCID50/mL) was verified by the Reed–Muench assay. A titrated viral suspension (105 TCID50/mL) was inactivated by heating at 65 °C for 30 min. The heat-treated viral suspension was tested by inoculating into Vero E6 cells for two passages to verify the absence of virus replication.

The viral suspensions were stored at −80 °C before the measurements were carried out. The protocol used for the preparation of the samples was the following:Application of 40 μL of poly-L-lysine onto a freshly cleaved mica piece (0.5 cm × 0.5 cm) previously attached to a circular metallic support (diameter of 18 mm) for the measurements in Jumping mode. For the QI^TM^ and correlative measurements the viral suspension was deposited onto clean glass coverslips (diameter 25 mm).Waiting for 15 min, then rinse thoroughly with ultrapure MilliQ water three times. After each rinse, a nitrogen gas flow was used to dry the surface.To disperse any aggregate, the Eppendorf containing the viral particles was vortexed five times, with each vortexing lasting for 5 s.Placement of 50 μL of the vortexed particles onto the mica surface.After another 15 min, addiction of 20 μL of TRIS buffer (5 mM) mixed with NaCl (150 nM) at pH 7.4, to maintain appropriate osmotic pressure and pH levels.Following 15 min, three rinses with the same buffer to remove all unattached virions and debris, and addiction of the right amount of liquid for the measurement (usually 40–50 μL).

For the experiments with Hyp, we prepared the Eppendorf with the virus by adding the PS so that the total concentration would have been 1 μM. We let it incubate in Eppendorf for 15 min. After this passage, we proceeded with the same protocol.

## 4. Conclusions

From our experiments, we have gained insight into the behavior of heat-inactivated SARS-CoV-2 viral particles under mechanical stress. The fatigue tests suggest that the virus possesses an extremely adaptable structure, consistent with characteristics expected for highly infectious viruses [22,23]. The results on Hyp loaded SARS-CoV-2 indicate significant changes in the virus’s stiffness, although the compensating effect of photooxidation remains unclear. The observed effects of Hyp on the mechanical properties in the absence of light may be of relevance for the reported dark-inactivation efficacy of the compound [6], for which the increase in membrane stiffness may result in a higher energy barrier for the viral–host membrane fusion [6]. The role of heat inactivation and sample variability currently prevents us from making conclusive statements about the effects of Hyp on the nanomechanical properties of viral particles. Therefore, additional data are needed to develop a more comprehensive understanding and mitigate the sample’s heterogeneity. Also, the use of less harmful active enveloped viruses is expected to provide more conclusive data on Hyp’s impact on viral mechanical properties. Nevertheless, we wish to emphasize that the preliminary data we present here are of relevance for a number of viral pathogens causing recurrent infectious diseases, such as membrane-enveloped viruses, which need fusion of viral and cell membranes for virus entry. Photoactive compounds able to target the viral envelope and affect the membrane fusion process by changing their mechanical properties may open new strategies in the development of broad-spectrum, multimodal antivirals [6], which could be less subject to the insurgence of resistance.

## Figures and Tables

**Figure 1 ijms-25-08724-f001:**
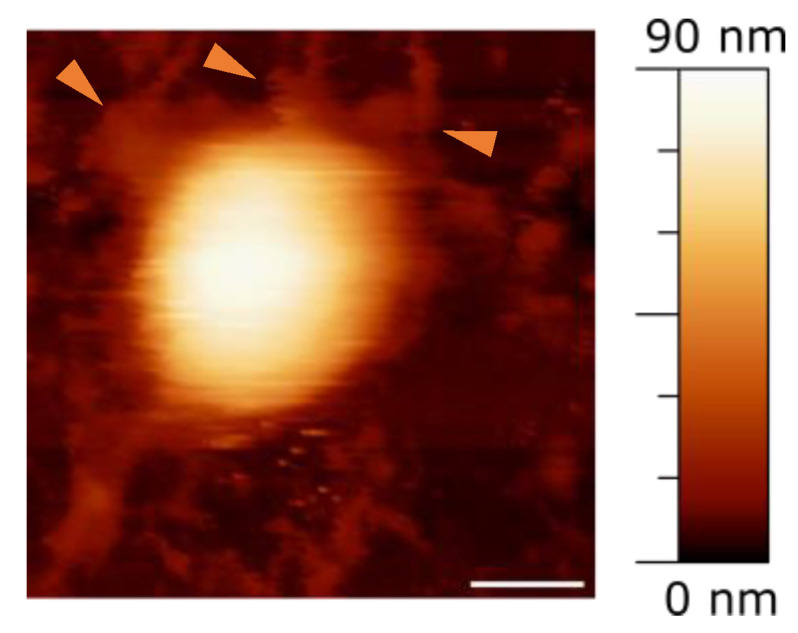
Image showing a complete viral particle on a mica surface. Orange arrows point to potential remnants of spike proteins. Scale bar = 50 nm.

**Figure 2 ijms-25-08724-f002:**
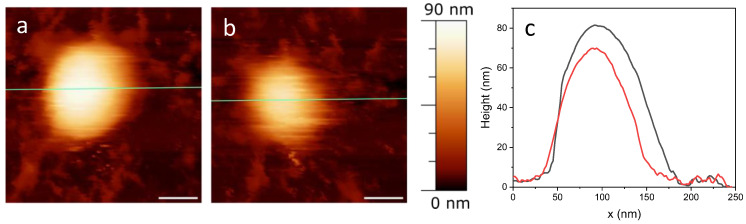
Representative fatigue experiment on SARS-CoV-2. The height profiles along the green line for the first (**a**) and the last (**b**) images are plotted in panel (**c**), black and red respectively. Scale bar = 50 nm.

**Figure 3 ijms-25-08724-f003:**
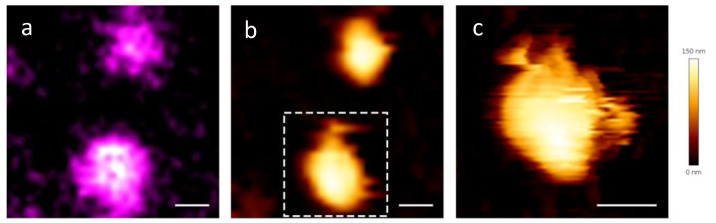
(**a**) Confocal and (**b**) AFM images of the same viral particles and (**c**) detailed image of the particle in the boxed area of panel b, that appears partially damaged. A Gaussian blur was applied to the confocal image, σ = 2. The AFM images were acquired in tapping mode. Scale bar = 200 nm.

**Figure 4 ijms-25-08724-f004:**
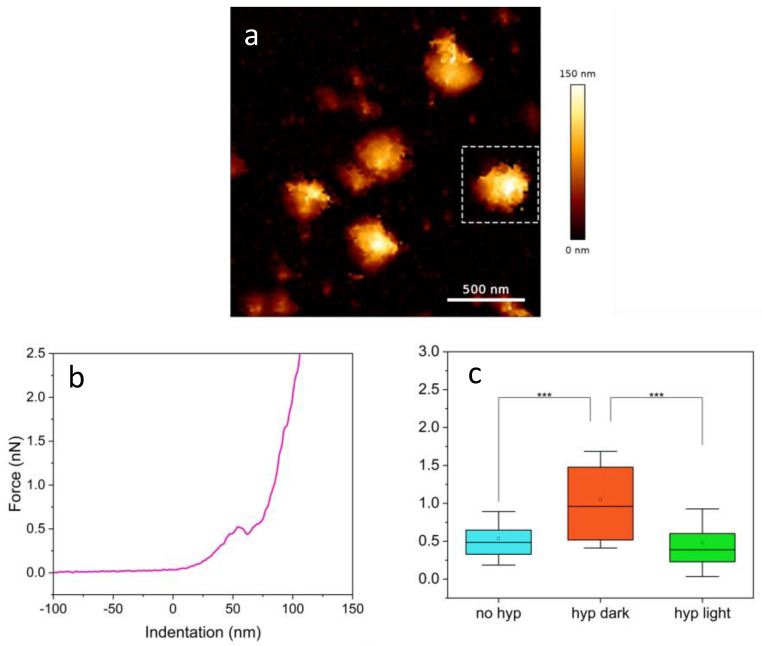
(**a**) QI^TM^ image of a few viral particles. A dashed square indicates the particle used for the breakthrough measurement reported in (**b**). Such graph displays the acquired force–distance curve showing visible penetration of the external viral envelope; (**c**) breakthrough force distributions under three different conditions with [Hyp] = 1 μM, asterisks indicate *p* value < 0.001.

**Table 1 ijms-25-08724-t001:** Summary of the results of the analysis of F–D curves shown in Figure 4c.

[Hyp]	N° of Curves	Breakthrough Force
0 μM	34	(0.54 ± 0.03) nN
1 μM in dark	71	(1.05 ± 0.06) nN
1 μM in light	36	(0.48 ± 0.04) nN

## Data Availability

Data is contained within the article.

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
