# Peer review of "Insights on the Mechanical Properties of SARS-CoV-2 Particles and the Effects of the Photosensitizer Hypericin"

_ijms, 2024, doi:10.3390/ijms25168724_

Round 1

Reviewer 1 Report

Comments and Suggestions for Authors

REVIEVER’S REPORT:

This research paper by Mariangeli and colleagues provides insights on the mechanical properties of SARS-CoV-2 particles and the effects of the photosensitizer Hypericin. ​ It explores the mechanical properties of heat-inactivated SARS-CoV-2 viral particles using Atomic Force Microscopy (AFM), revealing a flexible structure under external stress. ​ The impact of Hypericin on the mechanical properties of the virus is assessed, showing that it affects the mechanical stability of the viral envelope. ​ This paper also contains a list of references related to AFM and its applications in various fields, including studies on virus shells, cell mechanics, nanoscopy, and the mechanical properties of biomembranes. ​ It provides information about the sensitivity of QITM measurements, experimental evaluation of the spring constant, data acquisition and analysis methods, nanomechanics measurements, correlative microscopy techniques, virus preparation and inactivation. It is very essential to understand the effects of Hypericin on the nano-mechanical properties of SARS-CoV-2 particles. This is an interesting and important area of research. The paper is generally well written; however, the quality of the paper can be enhanced if the following points can be addressed.

1.     What specific techniques or methods were used to inactivate the SARS-CoV-2 viral particles for the AFM experiments performed in this study? Please describe them in detail in the materials section.

2.     How was the effect of Hypericin on the mechanical stability of the viral envelope measured?

3.     What were the key observations noticed regarding to the mechanical properties of the heat-inactivated SARS-CoV-2 viral particles?

4.     What were the specific findings or observations regarding the effects of Hypericin on the mechanical properties of the viral envelope? please discuss in the discussion.

5.      What were the specific experimental parameters and conditions used for the AFM measurements, such as the scanning speed, force setpoint, and tip type. Please mention them in the methods section?

6.     Did the authors face any challenges encountered during the AFM experiments during the study, and if so, how were they addressed? Please discuss them in the discussion section. Did the authors also tend to study any other factors or variables that could potentially influence the mechanical properties of the viral particles, such as temperature or humidity or other key factors?

7. What are the essential steps that need to be takes for clinical applications of Hypericin for COVID management.

Reviewer 2 Report

Comments and Suggestions for Authors

The reviewed work concerns the analysis of the influence of hypericin on SARS-CoV-2 virus particles. The Authors used Atomic Force Microscopy in their studies. Overall, the work is written correctly. The introduction contains the most important information about the problem, and the aim of the work is clearly and precisely formulated. In general, the methodology is described in detail, although there is a lack of information regarding the conditions of conducting the research "in light" and "in dark." The topic of the work and the research direction seem interesting from a methodological point of view. Nevertheless, I believe that the studies should be continued and expanded to obtain data that allow for a reliable presentation of conclusions and the full achievement of the work's objective. I suggest that the Authors thoroughly discuss the issue of the direct interaction of drugs with virus particles. It would also be worthwhile to better justify the undertaken research in terms of the applicability of the obtained results.

Comments on the Quality of English Language

Minor editing of English language required.

Round 2

Reviewer 2 Report

Comments and Suggestions for Authors

The Authors have addressed the questions and revised the manuscript according to the suggestions. At this stage, I have no further comments regarding the presented results. However, I believe that, considering the title, objectives, and outcomes, the goal of the study has been only partially achieved. Further research in this direction is necessary.

Comments on the Quality of English Language

Minor editing of English language required.